# Assessment of Anchors Constellation Features in RSSI-Based Indoor Positioning Systems for Smart Environments

**Alessandro Cidronali**, **Giovanni Collodi** *, **Matteo Lucarelli**, **Stefano Maddio**, **Marco Passafiume** and **Giuseppe Pelosi**

Department of Information Engineering (DINFO), University of Florence, 50139 Florence, Italy; alessandro.cidronali@unifi.it (A.C.); matteo.lucarelli@unifi.it (M.L.); stefano.maddio@unifi.it (S.M.); marco.passafiume@unifi.it (M.P.); giuseppe.pelosi@unifi.it (G.P.)
* Correspondence: giovanni.collodi@unifi.it; Tel.: +39-055-2758544

**Abstract:** In this paper, we assess the features of a rectangular constellation of four anchors on the position estimation accuracy of a mobile tag, operating under the IEEE 802.15.4 specifications. Each anchor implements a smart antenna with eight switched beams, which is capable to collect Received Signal Strength Indicator (RSSI) data, exploited to estimate the mobile tag position within a room. We also aim at suggesting a deployment criterion, providing the discussion of the best trade-off between system complexity and positioning accuracy. The assessment validation was conducted experimentally by implementing anchor constellations with different mesh sizes in the same room. Mean accuracies spanning from 0.32 m to 0.7 m on a whole 7.5 m × 6 m room were found by varying the mesh area from 1.19 m$^2$ to 17 m$^2$, respectively.

**Keywords:** wireless positioning; smart environments; distributed networks; collaborative systems; practical deployment; wireless communications; Internet-of-Things

---

## 1. Introduction

It is widely recognized that user localization in Global Navigation Satellite Systems (GNSS) denied environments represents an essential feature for the capillary diffusion of the paradigm [1–3]. This concept, which has recently been extended and referred to as the "Internet of Things" (IoT), consists of a complex system where physical objects are provided with sensors and actuators, as well as network connectivity. This paradigm relies both on the devices low-energy demand, and their ability to reach an information network where they can contribute to distributed processes [4–7]. These elements typically communicate information about their status as well as the state of the surrounding environment. In many cases it is required for the transmitted data to be associated with the context in which the mobile agent operates to improve the information effectiveness, in particular to be linked to its localization [8]. The position needs to be seamlessly available even in those environments with no access to GNSS. The need for localization, as well as tracking, becomes relevant not only from the network management point of view but also to improve the capability of Wireless Sensor Network (WSN) systems in a smart home environment [9,10]. Nevertheless, enabling or creating a localization system with an IoT network is not a trivial task. The reason is that low cost, compactness and long lasting autonomy are mandatory for IoT devices to enable the most widespread diffusion. This fact results in objects with very limited hardware components, and very stringent specifications for elements such as CPU, memory and battery, [11]. It is commonly agreed that this problem is not yet solved. In fact, a technology (or a combination of technologies) does not seem to be available that is capable of recreating in indoor environments the experience that GNSS offers for outdoor spaces.

In this work, the problem is addressed by following the paradigm of increasing the amount of information exchanged by a number of wireless nodes, while assuming a limited use of additional ad-hoc hardware to that already implemented in the established wireless infrastructure. The reason is that positioning should be offered as an application layer on top of the data networking layers, with a minimal impact on the already existing networks.

In this context, several positioning approaches relying on wireless infrastructures based on IEEE 802.11x/802.15.4, have been presented in recent literature [12,13]. The adoption of commercial radio transceivers makes the use of the Received Signal Strength Indicator (RSSI) a common approach. Typically, an RSSI enables fingerprinting, which has its main drawback in its requirement of a periodical and lengthy on-site calibration phase, to overcome environment characterization uncertainties. Alternatively, a dense distribution of specific beacons can be used. This type of solutions is not feasible in IoT environments, especially in typical smart environment applications, which include integrations of positioning systems and autonomous machines. For instance, in smart home environments a robot can use data from a positioning system to improve the context awareness of interacting objects, and not only to move into the specific scenario [14]. This requires a localization system characterized by low cost, low power, ease-of-installation, and ease-of-maintenance.

In [15] and [16], the authors proposed an indoor localization system based on a mesh network of IEEE 802.15.4 compliant anchor nodes. In those works, as well as hereinafter, the mesh is associated to the location of the positioning system anchors, which define the mesh vertexes. The technology is based on the installation of a set of switched anchor nodes equipped with Switched Beam Antennas (SBA) [15–17], each one capable of IEEE 802.15.4 communications. After having installed a proper number of anchors working together as a "distributed" router, the network is able to offer wireless connectivity to its nodes, while collecting a long set of distributed RSSI values. The various received signal power levels obtained with this procedure are then exploited using proper algorithms to obtain a position estimation, based only on received signals magnitude (i.e., phase-less). A high level of accuracy is normally achieved after an on-site calibration phase (as opposed to [16]), or additional tracking based on odometric readings (as opposed to [13,18]). Using only RSSI (as defined in the IEEE 802.11x and 802.15.4 standards) readings as input data for positioning algorithms, this method is suitable to any IEEE compliant wireless communication protocol, making its implementation in any pre-existent wireless network infrastructure easily achievable (i.e., [19,20]).

In this paper we further extend the concept already presented in [15,16], by introducing and validating a comprehensive assessment about the trade-off between system positioning complexity and positioning accuracy in a typical office room site.

The paper is organized as it follows. In Section 2 we review the architecture of the positioning system, including that of the mobile tag. The distributed positioning technique is reviewed in Section 3, which includes also the analysis of the anchor constellation shape and size. The results of this analysis are discussed in details in Section 4, also comparing simulation and experimental data related to a real test site. A discussion about the outcomes of the work concludes the paper.

## 2. Localization System Architecture

### 2.1. Front-End Architecture

As shown in [16], the localization system is based on a constellation of anchors capable of exchanging communication data with the target node, while simultaneously collecting the corresponding set of RSSI.

A block diagram of the anchor system architecture is given in Figure 1 [15]. It is based on an 8-channel SBA, a SP-8T microwave switch, and a System-on-Chip (SoC), which implements a radio-frequency transceiver and a microcontroller unit (MCU). The hardware scheme proposed in Figure 1 is deliberately simplified to highlight the ease of implementation of the anchor architecture. The vast majority of Commercial Off The Shelf (COTS) SoCs are suitable to provide the required features

for this applications. The device chosen for this work is the Texas Instruments CC2530, which includes a 2.45 GHz transceiver supporting the IEEE 802.15.4 protocols and integrates a 8051 MCU with 8-kB of RAM. The additional feature required to the MCU consists of the calculation and the transmission of the Received Signal Strength Indicator (RSSI) to the localization engine. In this project the RSSI is transmitted by the Ethernet interface. The MCU provides a set of GPIOs, used also to control the single-pole eight-trough (SP-8T) RF switch that sequentially routes the transmitted or received signal to each of the antenna elements of the SBA. In this project, we adopted a HMC321, which is a low-loss SP-8T non-reflective switch capable to work in the DC-8 GHz bandwidth with less than 2.5 dB losses at the operative frequency. This bandwidth is much larger than what is required for this application; this is because this architecture is extremely flexible and modular, and the switch board has been used also for other experiments at different bandwidths. The firmware provides the proper synchronization between the RSSI data collection and the switch settling time, which is 150 ns, thus granting a bijective correspondence between the receiving antenna and the obtained RSSI by avoiding the spreading of a single packet through multiple antennas. An RSSI collection cycle is completed when a network packet is exchanged with the mobile node for each of the 8 antennas of a single SBA. In the case of the IEEE 802.15.4 radio access, the data transfer rate is about 250 kbit/s with a packet length of 60 bytes. The entire set of RSSI data for each anchor is collected in about 13.3 ms. An additional time has to be added to this value for the position algorithm execution, which is around 10 ms depending on the chosen algorithm and the power of the machine on which it is performed.

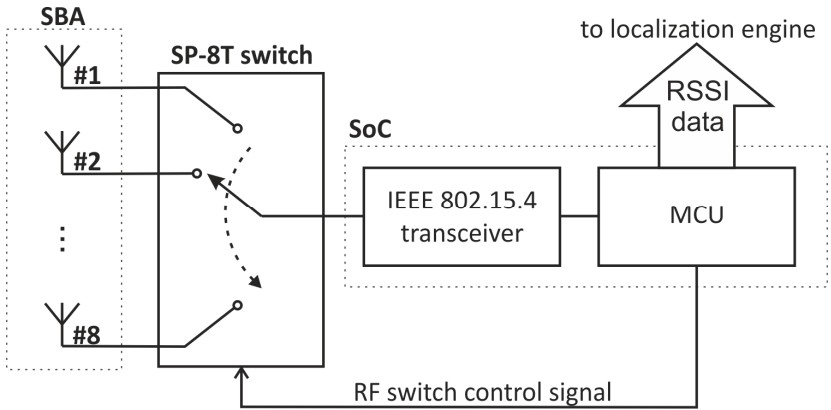

**Figure 1.** Anchor system architecture.

An entire RSSI data set is collected at the receiving of 8 packets by each anchor node. The anchor network is able to optimize the packet flow by synchronizing packet readings between different anchor nodes, thus the mobile nodes have to transmit only 8 packets. Referring to CC2530 transceiver datasheet, when powered at 3 V supply voltage, the current absorption during active transmission is about 29 mA corresponding to a peak power consumption of 87 mW, while during radio receiving the current absorption is 24 mA (with a peak power consumption of 72 mW).

According to the general need about which, mobile nodes transmission profile must be tailored to maximize battery life, they send their data as a burst of 8 repetitions of data packets, only at the required localization refresh according the radio channel is sensed to be free. Following this, their average power consumption decreases, while the localization refresh time increases. As an example, considering a burst of 8 packets, equal to 480 bytes or 14 milliseconds of transmission time, a refresh time of 1 second leads to an average power consumption of 1.2 mW.

Because the anchor nodes are part of fixed network infrastructure, to deal with user nodes random packet transmissions, they must be always ready to receive data, so they are always in receiver enabled state. By this, their average power consumption exactly equals peak consumption of 72 mW.

## 2.2. Switched Beam Antenna Architecture

The key part of the anchor system architecture is the SBA, which is based on an array of seven antennas, arranged with one hexagon at the SBA center and six perimetric pentagons. Figure 2 shows the mechanical arrangement of the SBA: it is a hemispherical structure with a maximum diameter around 146 mm, a height of 53.8 mm and with a dihedral angle of 120 degree.

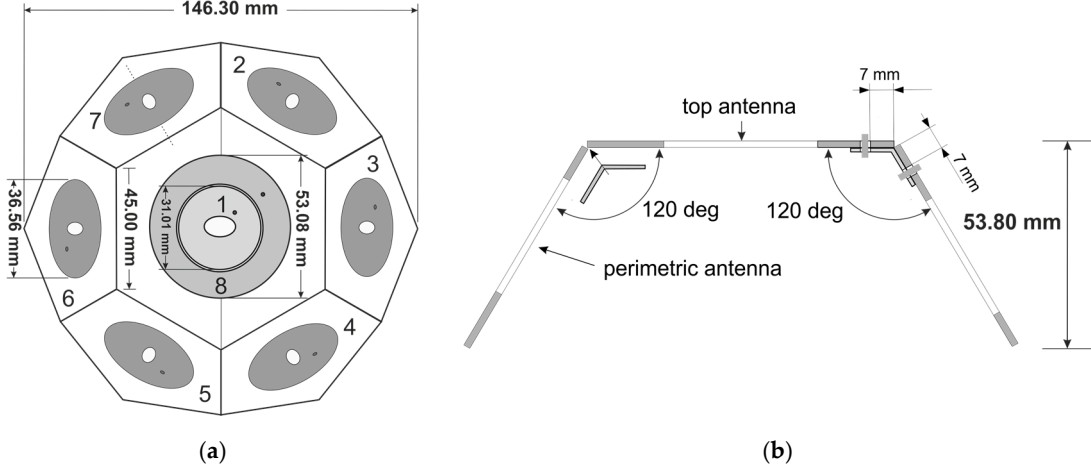

| (a) | (b) |

**Figure 2.** The mechanical structure of the Switched Beam Antenna; (**a**): top view, (**b**): side view.

The dihedral angle is a design parameter depending upon the operative conditions, in particular coverage area, and noise, as discussed in [21], nevertheless its optimum value is between 100 and 130 degree for circular coverage area radius up to 2.5 m. The SBA exhibits six identical perimetric antenna, while the top antenna differentiates from the previous ones as it admits two radiation modes and corresponding input ports, [22]; the effective beams are therefore 8 in total. Each of the perimetric elements is a Circularly Polarized (CP) antenna designed by the principle explained in [23], which is basically a technique capable to generate two propagation modes with a single feeding signal, by splitting it in two orthogonal and properly phased components. This is obtained by removing the conductor in an elliptical region centered within the disk. The top antenna is instead a dual mode circularly polarized antenna capable to provide a regular cardioid pattern with the maximum of radiation in the boresight exploiting its first input pin, along with a second patch that provides a toroidal patter with the maximum radiation at 45 degree from the antenna axis obtained using its second pin.

The adoption of circular polarization is motivated by two main reasons: the first consists in the need to reduce any signal fading due to cross-polarization coming from a generic target–anchor orientation. The second one consists in the inherent robustness of this propagation mode to multi-path phenomena. The SBA patch antennas are capable of operating in the 2.4–2.48 GHz band.

In Figure 3 the picture of the prototype adopted in this work is reported. The pictures show the SBA assembly (Figure 3a), with the top antenna and 3 out of the 6 perimetric antennas. The top antenna implements a number of parasitic elements that improve the gain of antenna element #8, the one with a toroidal pattern, which results particularly affected by the global arrangement. The rear view visible in Figure 3b shows the control unit of the SBA and the other parts of the anchor architecture shown in Figure 1. This specific SBA is implemented in each anchor node by using a multiplexer properly synchronized with the RSSI block; this grants that the RSSI acquisition is related to the specific array element and thus to a specific space sector. The multiplexing between the antennas operates in a synchronized mode for all the anchors belonging to the same constellation to avoid signal clashes.

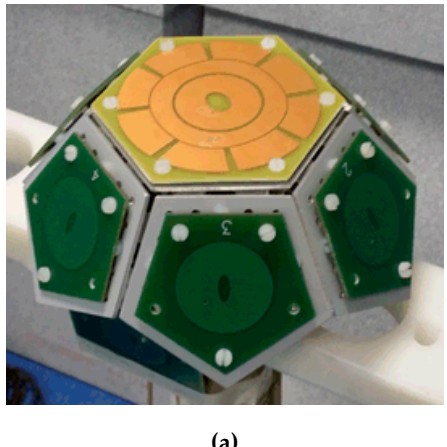

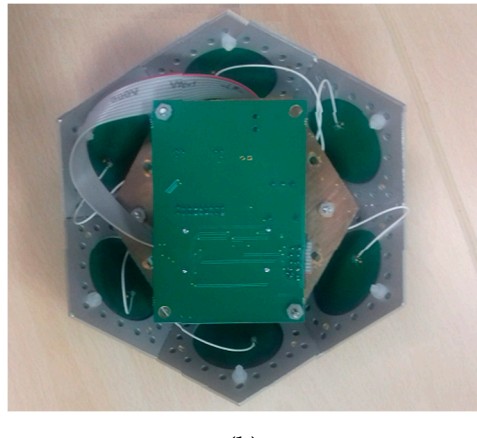

(a)                                                                 (b)

**Figure 3.** Switched Beam Antenna prototype: (**a**) upper view; the top antenna in this realization presents some parasitic elements to improve the radiation gain of antenna #8; (**b**) rear view, with the control unit board in view.

The radiation pattern of the SBA is shown in Figure 4, in the azimuthal plane ($\phi$) and elevation ($\theta$). In particular, Figure 4a shows the measured radiation patterns for all the 8 antennas at elevation angle $\theta = 90$ degree. This is not the boresight direction for antennas 2–7, thus they exhibit maxima lower than their absolute maximum gain. From the graphs it is possible to see that the respective direction of maximum radiation is uniformly spread across the radial direction. In this picture, antennas 1 and 8 are observed by their horizontal angle of view, far from their maximum, and thus do not provide any pattern variation. Figure 4b reports the corresponding radiation patterns for each antenna in the elevation plane; in this case all the antennas are observed at different angle of view and thus provide gain variation. In particular, the complementarity of the antenna 1 and 8 gain patterns is clearly observable, thus providing an additional piece of information to the localization engine. Antenna 8 exhibits a lower gain with respect to the others because it is affected by the presence of antenna 1; the outer parasitic elements partly compensate this effect. Because the specific selection of the $\theta$ angle, in this graph the radiation patterns are very similar as the elevation plane contains the dihedron formed by these two antenna.

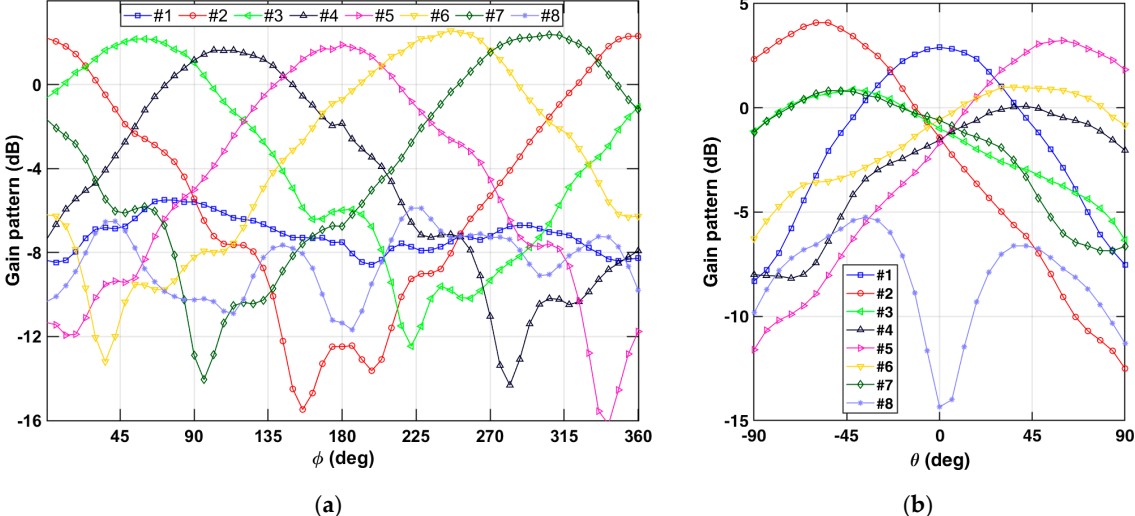

(a)                                                                 (b)

**Figure 4.** Switched Beam Array: (**a**) measured pattern in azimuthal plan; (**b**) pattern at $\phi = 70°$ (side pointing perpendicular direction).

The global pattern results from the sequence of each cardioid pattern, and provides Spatial Division Multiplexing Access (SDMA). A single anchor node is hence capable of performing signal direction-of-arrival estimation, [23], and is inherently suitable for an integration in more complex anchors constellations distributed across an entire area.

### 2.3. Mobile Tag Architecture

The mobile tag subject to the positioning shares its architecture with the anchor node with the exception of the ethernet connection, the SBA and its associated switch. The power supply is provided by two 1.5 V AA batteries. In this work, the SoC is connected with a planar CP patch antenna, as shown in Figure 5a. The radiation pattern is provided in Figure 5b. The radiation pattern of the target node is characterized by a main radiation lobe directed along the antenna axis and exhibits a sufficiently regular behavior around it, [20]. This shape resembles that of a typical Planar Inverted F-Antenna (PIFA), as implemented in generic mobile terminals, [24]. In fact, PIFA antennas cannot be considered to have "pure" radiation patterns, as these devices always show an upward oriented non-isotropic pattern when considering a typical handset use case. In addition, the PIFA is not suitable for CP which is instead preferable for this application to face the severe multipath effects that can be present in many realistic scenarios. More elaborated mobile tag antenna designs can be taken into account, [25], nevertheless they require a complex feeding network and in many circumstances result not compatible with the applicative scenario. For instance, this constraint is undesirable for a compact and low cost device such as the proposed tag node and have therefore not been used in this work.

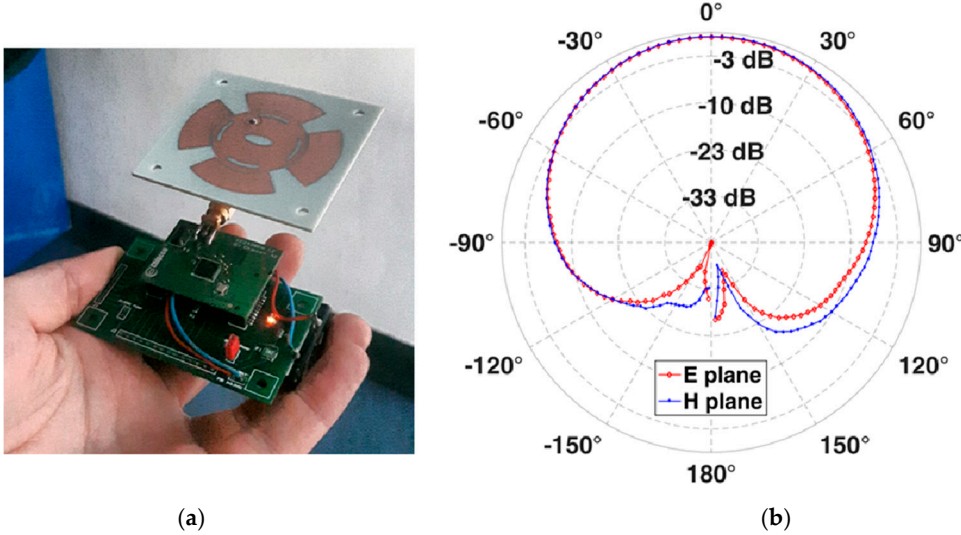

(**a**)　　　　　　　　　　　　　　　　　　　　　　　　(**b**)

**Figure 5.** Experimental target node (**a**) and its radiation pattern (**b**).

## 3. Distributed Positioning

### 3.1. The Position Estimation

The numerical estimation of the mobile tag position is based on signal magnitude level measurements, and is generally pursued by minimizing an objective function. Typical objective functions are expressed by sums of squared residuals, likelihood functions, posterior density functions, risk functions, as well as robust loss functions, with the two most common being the Least Squares (LS) and the Maximum Likelihood (ML) functions [1].

In this paper, we focus on an algorithm based on a derivation of the ML method, whose estimation only requires the assumption of a first order knowledge of the range measurement probability distribution function. It is based on the minimization of the calculated variance for both measured and expected data from an unknown position, [16]. Its operating principle assumes the acquisition of a

coherent set of signal levels during network communication between the mobile tag and a generic number, N, of available anchors. The collected data consist of a steering vector, containing all the RSSI values obtained from each antenna element of each anchor composing the constellation; the steering vector is defined as:

$$\acute{S} = \begin{pmatrix} \acute{S_1} \\ \acute{S_2} \\ \vdots \\ \acute{S_N} \end{pmatrix},$$ (1)

where $\acute{S_i}$ denotes the steering vector of the *i*-th anchor, with:

$$\acute{S_i} = \begin{pmatrix} s_{1i} \\ s_{2i} \\ \vdots \\ s_{Mi} \end{pmatrix},$$ (2)

and where $s_{ki}$ is the RSSI collected by the *k*-th antenna of the *i*-th anchor, with *M* the number of SBA antenna elements. The positioning technique adopted in this work is hereinafter described, considering the *i*-th steering vector composed by the RSSI data obtained from the *M* antenna elements composing the single anchor node. The position estimation relies on the assumption of a known mobile tag height $z_{TAG}$ (*z*-axis), used to define the steering vector reference map, $M(x,y)$. The latter is obtained as the projection of the SBA patterns throughout the (*x*,*y*) localization domain calculated by a first-order propagation model, [16], between a generic tag position (*x*,*y*) and the anchor known position. This is characterized by the path-loss model $20 \log \left( \frac{D_i}{\lambda_0} \right)$, with $D_i$ the distance between the mobile tag at (*x*,*y*) and the *i*-th anchor node, and $\lambda_0$ the free-space wave length at the operative frequency. Furthermore, it is assumed that between the anchors and the mobile tag persists a line-of sight. Under these realistic hypothesis, the tag position estimation $(\hat{x}, \hat{y})$ is obtained by minimizing the variance of the difference between the steering vector and the expected signal map vector across the entire analysis domain:

$$(\hat{x}, \hat{y}) = \text{argmin}_{(x,y)} C(x, y),$$ (3)

with

$$C(x, y) = \text{var}_i \left\{ \acute{S} - M(x, y) \right\}.$$ (4)

In Equation (4), the variance of the difference between the steering vector and the expected signal map for the actual position is given by:

$$C(x, y) = \frac{1}{MN} \sum_{i=1}^{MN} \left[ (s_i - m_i) - \frac{1}{MN} \sum_{j=1}^{MN} (s_j - m_j) \right]^2$$ (5)

As demonstrated in [16], the robustness of the estimator in Equation (5) is able to counterbalance the approximation introduced by the map model and the large variability of possible user node heights.

### 3.2. Comparison between Triangular and Squared Shapes and Parameters

An a priori analysis of the anchor constellation shapes suitable for a proper coverage of the positioning domain can be effectively obtained by the results of the information theory. Extracting the inverse of the measurement covariance matrix, **F**, as the weight matrix, the variance leads to the Cramer–Rao lower Bound (CRB) for an unbiased estimator, E. We can write that [26],

$$var[E] \geq \text{CRB}(x, y) = \text{CRB}_x(x, y) + \text{CRB}_y(x, y),$$ (6)

where

$$\text{CRB}_x(x,y) = \sigma^2 \frac{\sum_{n=1}^{NM}\left[\frac{\partial G_n}{\partial x}\right]^2}{det(\mathbf{F})}; \quad \text{CRB}_y(x,y) = \sigma^2 \frac{\sum_{n=1}^{NM}\left[\frac{\partial G_n}{\partial y}\right]^2}{det(\mathbf{F})} \tag{7}$$

In Equation (7) $G_n$ represents the projection on the $(x,y)$ domain of the $n$-th antenna element radiation pattern of the SBA and $\sigma^2$ the variance of the observed data. The CRB represents the accuracy limit of the positioning estimator with respect to its several parameters, which in this work case are the shape of the constellation (i.e., triangular versus rectangular), their size, and the additive white Gaussian noise characterized by its variance $\sigma^2$. The Figure 6 illustrates the CRB calculated in a 10 m × 10 m square domain, corresponding to a constellation of three anchors arranged in an equilateral triangular shape, with various triangle edge size, and $\sigma = 1$. The domain corresponds to the empty space. From the picture we can see that the minimum error is expected in correspondence of the anchors and reducing the edge size some kind of fusion of the minima occurs, leading to a unique circular region of minimum around the center of the domain. From the graphs we see that the triangular mesh can return an average accuracy across the entire domain which is almost flat up to 7 m, while within the shape the average CRB increase with the square of the shape size. In Figure 7 we report a similar analysis considering a square shape constellation. In this case we can see a behavior qualitatively similar to the previous one but with a more uniform distribution within the entire domain. Figure 8 reports the accuracy in terms of CRB for both cases. From this we can observe that the rectangular mesh leads to better average values for the accuracy for the ideal mobile tag position estimation, being always better than the corresponding triangular mesh area. Given this, the article will focus on the rectangular shape, although it requires an additional anchor with respect to the triangular case.

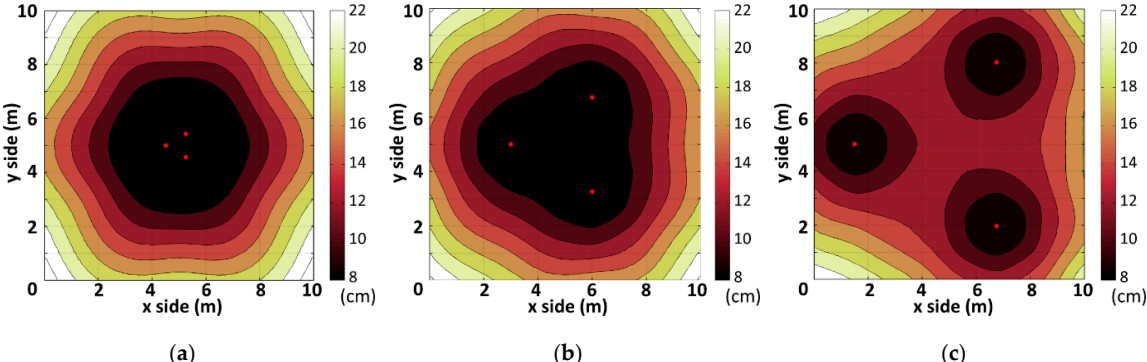

(a)          (b)          (c)

**Figure 6.** Cramer–Rao lower Bound (CRB) evaluation for triangular mesh at different edge sizes: (**a**) 0.85 m; (**b**) 3.45 m; (**c**) 6 m.

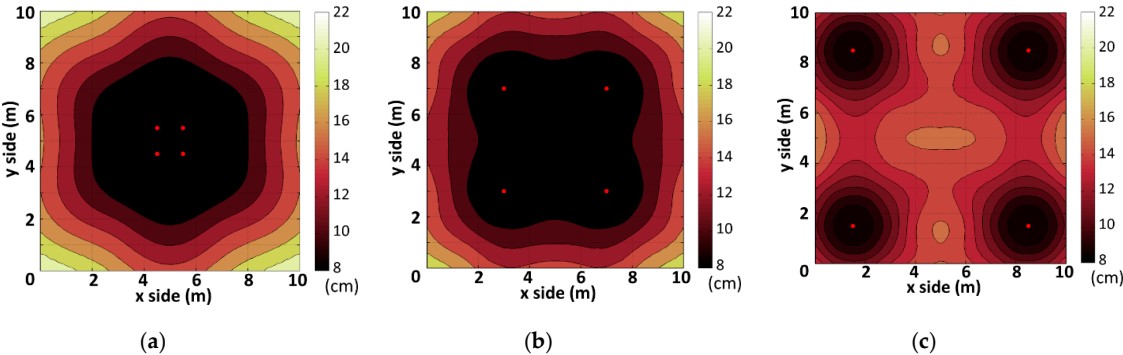

(a)          (b)          (c)

**Figure 7.** CRB evaluation for rectangular mesh at different edge sizes: (**a**) 1 m; (**b**) 4 m; (**c**) 7 m.

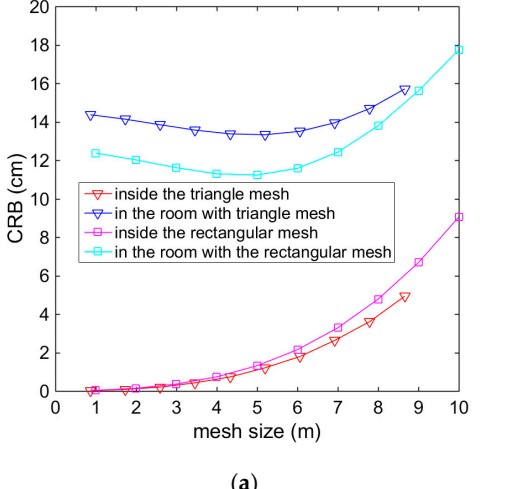
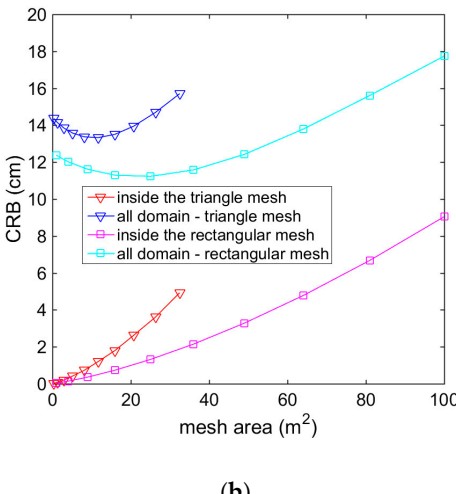

(**a**)                                                    (**b**)

**Figure 8.** Calculation of the average CRB for different mesh shapes and corresponding sizes, as a function of (**a**) the mesh edge size, and (**b**) the mesh area; $\sigma^2 = 1$.

## 4. Impact of the Constellation Mesh Area on Localization Accuracy

This section analyzes the effects of the mesh area on the localization accuracy and error coverage of the proposed system within a real case indoor environment. On the basis of the previous Section results, we focus on a rectangular constellation with 4 identical anchors at its vertexes. We experimentally verify the accuracy of the positioning by three mesh size cases. For all the analyzed cases, the position estimation is carried out by the estimator in Equation (3). The analysis compares the experimental data with simulations.

### 4.1. Experimental Site

The scenario under consideration is a typical small office environment with a total area of 7.5 m by 6 m, for a total area of 45 m$^2$, with the usual furniture. Its map is reported in Figure 9a, and it is possible to see that it includes furniture, metallic objects and office closets spread around the area. The tests were performed while allowing the office employees to continue their habitual tasks. The latter condition results to affect the variance of the observed data, $\sigma^2$, defined in the Equation (7). The picture of the test site is reported in Figure 9c where we observe the single constellation rectangular mesh installed on the ceiling, with the three out four anchors visible, and the mobile tag installed on a tripod. This installation guaranties a line-of-sight between the anchors and the mobile tag, which is a requirement for a proper positioning estimation.

The main goal of this work consists on the experimental investigation of the relation between the rectangular mesh size and the position estimation accuracy, thus making the anchor nodes location a degree of freedom to design the network. In general, a system architect designer can consider other degrees of freedom like the number of anchors and their arrangements in different mesh shapes, being the latter having already been considered by an idealized point of view in the previous Section 3.2.

To fully evaluate the effects of the anchor distribution on the localization accuracy, the mobile tag position has been estimated on a large number of points spread all over the room; these are visible in Figure 9a. In particular, the grid is composed by 299 points regularly spread throughout the room, wherever it was possible compatibly with the obstacles present in the environment, and spaced by 0.5 m × 0.5 m both by row and column, with points on adjacent columns vertically shifted by 0.25 m. The three test anchor constellations are visible in Figure 9b, they are characterized by different edge thus area sizes, namely a "large", a "medium" and a "tiny" mesh configurations; Table 1 summarizes the constellation parameters for each case. Overall anchors were hanged at the ceiling, at height $Z_i = 2.8$ m.

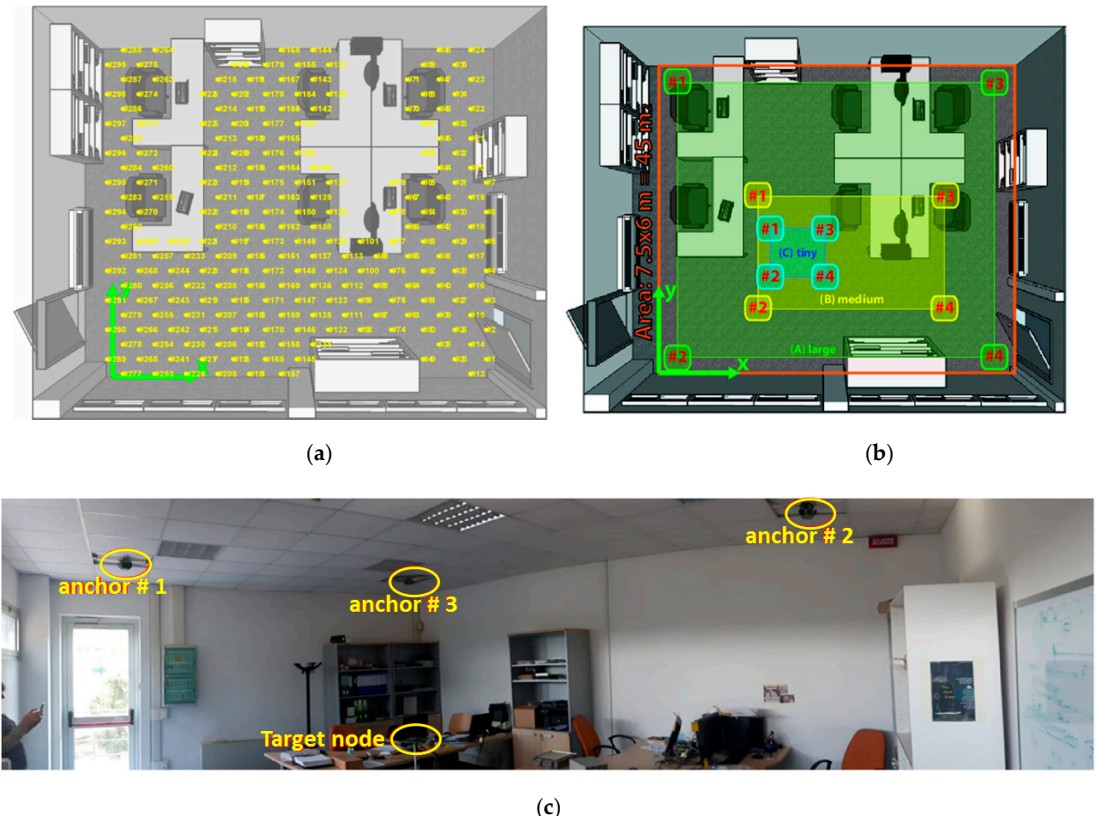

**Figure 9.** (**a**) Bidimensional map of the experimental test site, (**b**) with the superimpose of the three constellations adopted in this work, and (**c**) the corresponding picture.

**Table 1.** Anchor–mesh configurations.

| Configuration | Anchor Positions | Mesh Area |
|---|---|---|
| (A) Large | #1 (1.50, 5.00) m; #2 (1.50, 1.00) m #3 (6.00, 5.00) m; #4 (6.00, 1.00) m | 17 m$^2$ 40% of room area |
| (B) Medium | #1 (2.20, 3.50) m; #2 (2.20, 1.20) m #3 (5.50, 3.50) m; #4 (5.50, 1.20) m | 7.6 m$^2$ 17% of room area |
| (C) Tiny | #1 (2.70, 2.65) m; #2 (2.70, 1.70) m #3 (3.90, 2.65) m; #4 (3.90, 1.70) m | 1.19 m$^2$ 2.6% of room area |

*4.2. Simulation Results*

In this work, the accuracy of the position estimator is firstly evaluated by a set of simulations. This permits to generate a test bench by which evaluate the results, and in prospective provide a model for the localization estimator in a given environment. This eventually permits to plan an effective anchor network topology in an optimal and reliable way, avoiding a lengthy on-site verification.

Given the fact that the selected localization algorithm relies on RSSI measurements as defined for IEEE 802.11x and IEEE802.15.4 wireless network standards, the synthetic localization error based on CRB evaluation provide a first order approximation of the obtainable results, not always reliable yet necessary the initial stage to define the real distributed positioning anchors network. This is due to the inherent formulation of the CRB, cf. Section 3.2, that provides the analytical characterization of the localization error assuming the steering vectors composed of the received signals at the antenna section. In [12], it is discussed how the RSSI can be considered as an output variable of a complex digital correlation algorithm with a high process gain. Given this, the effective system performance should be analyzed through an algorithm capable of encompassing every data processing layer, [26]. In particular, the RSSI noise model should be described using its own detection error parameters rather

than the effective physical signal-to-noise ratios: such detection error parameters are defined within protocol standards, and they depict a less noisy measurement scenario than the straight physical one [27]. The RSSI noise model can be brought back to an Additive White Gaussian Noise (AWGN) model, as described in [18]. The simulation process applies the localization estimator described in Equation (3), by considering the following steering vector in the (x,y) domain:

$$\acute{S} = \langle s_{11} \vee s_{11} \vee \ldots \vee s_{MN} \rangle^H = \mathrm{M}(x, y) + \acute{T} + \acute{N}, \tag{8}$$

where the steering vector $\acute{S}$ length is $\sum\limits_{i=1}^{N} M_i = MN$, while $\acute{T}$ contains the radiation pattern of the mobile tag node observed by the N anchors:

$$\acute{T} = \begin{pmatrix} \acute{T}_1 \\ \acute{T}_2 \\ \vdots \\ \acute{T}_N \end{pmatrix} = \begin{pmatrix} \mathrm{G}_{\mathrm{TAG}}(\vartheta_1, \varphi_1) \cdot I_{M_1,1} \\ \mathrm{G}_{\mathrm{TAG}}(\vartheta_2, \varphi_2) \cdot I_{M_2,1} \\ \vdots \\ \mathrm{G}_{\mathrm{TAG}}(\vartheta_N, \varphi_N) \cdot I_{M_N,1} \end{pmatrix}. \tag{9}$$

The term $\mathrm{G}_{\mathrm{TAG}}(\vartheta_1, \varphi_1)$ is the mobile tag antenna gain seen by each anchor antenna element. The anchor and the mobile tag are considered as points in space. The unknown mobile tag spherical coordinate $(\varphi_i, \vartheta_i)$, as observed by each anchor node installed on the ceiling at $(x_i, y_i)$, are:

$$\begin{cases} \varphi_i = \mathrm{atan}\left(\frac{y-y_i}{x-x_i}\right) \\ \vartheta_i = \frac{\pi}{2} - \mathrm{atan}\left(\frac{z_i-z}{\sqrt{(x-x_i)^2+(y-y_i)^2}}\right) \end{cases}, \tag{10}$$

while the noise contribution $\acute{N}$ is described by:

$$\acute{N} = N_{MN}(\mu, \sigma) = N_{MN}(0, \varepsilon_{RSSIMAX}). \tag{11}$$

Here it is assumed an RSSI noise characterized by a Gaussian model with null mean and variance $\varepsilon_{RSSIMAX}$, [16]. Moving on to the expected RSSI data, the measurement noise term embeds both the effective RSSI detection error and second order effects (e.g., fast fading). It is possible to get a reliable characterization of the RSSI variance, removing fast-fading by the rejection techniques described in [12]. Considering an example of the typical recorded RSSI data by the anchor SBA elements 1–7, reported in Figure 10, the worst cases show a maximum RSSI drift of about 5 dB during a single steering vector acquisition. Given this observation, we can set for the simulations the value $\sigma = \varepsilon_{RSSIMAX} = 5$. Note that in a real implementation, the RSSI variance can be reduced by applying a mobile mean along the acquired steering vectors.

The simulation results across the entire area for the three cases of study are reported in Figure 11, in terms of Cumulative Error Distribution Function (CDF) for the area within the single rectangular mesh and across the entire room, respectively. In this set of simulations, we generated 5 RSSI randomly distributed according the noise model described above at each of the 299 points distributed as in Figure 9a. From the figure we can observe that, due to the reduced dimension, the tiny mesh is capable to estimate with comparably smaller error the position within the area defined by the mesh itself, but it is not suitable for the outer part, as it determines high errors at the border of the room; this is already predicted by the investigation reported in Section 3.2 and specifically Figure 7a. For the other mesh sizes, the data of Figure 11 show a convergence between the outer and inner mesh results, thus confirming that the large mesh provides almost the same results within and across the entire area, as the two results almost the same.

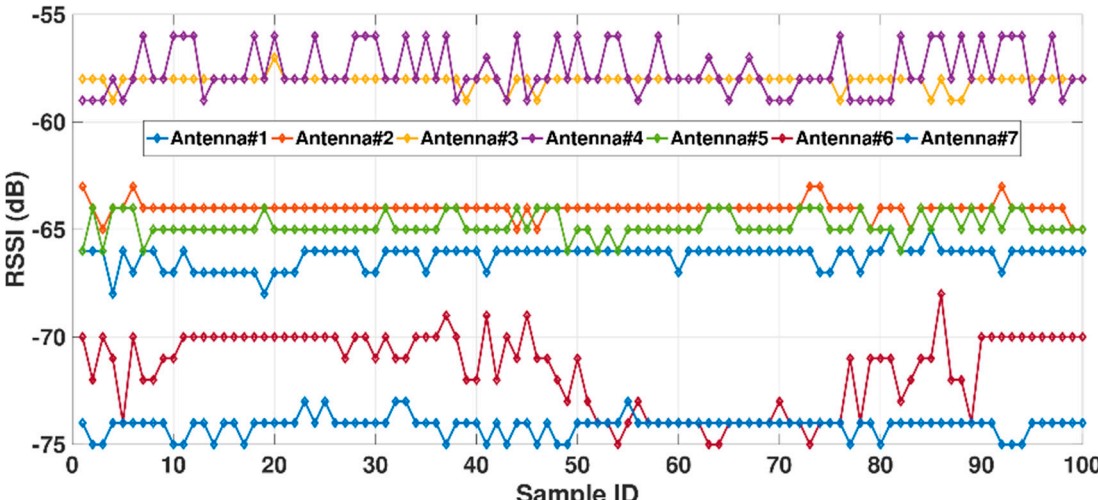

**Figure 10.** Typical Received Signal Strength Indicator (RSSI) acquisition trace: case C, anchor installed at room up-right corner (5.75, 5.00) m.

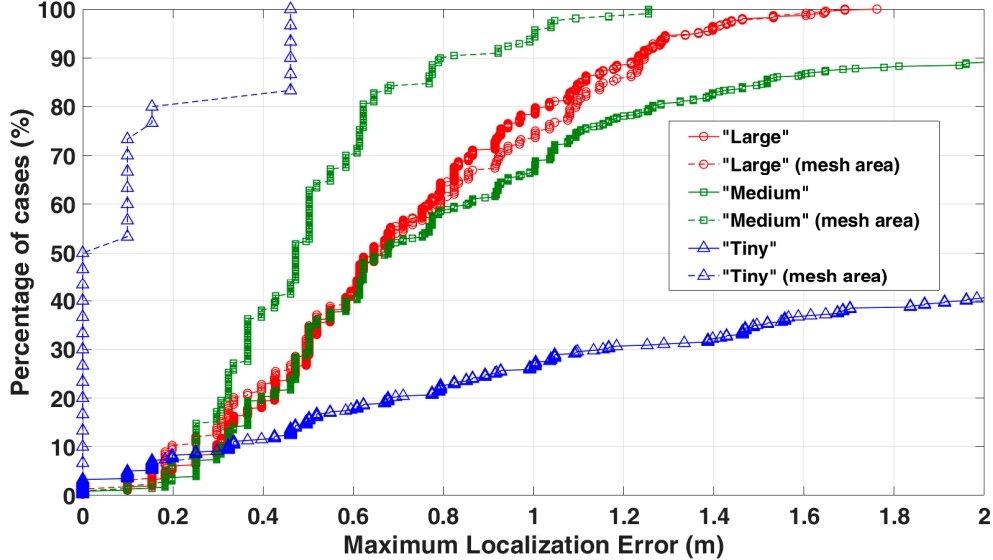

**Figure 11.** Simulated localization error CDFs, with the three anchor networks. Data acquired at the 299 points in the room of Figure 9a, assuming the mobile tag pointing upward, 5 averaged RSSI data for point and each antenna element, mobile tag height = 1.1 m.

For a better comparison of the data, the results are summarized in Table 2. They clearly show that a trade-off involved in the anchor distribution exists: a wider mesh increases the sub-metrical coverage on the full area, while smaller configurations increase the precision inside the mesh yet degrade significantly the overall area performance. With this conclusion we have an objective way to evaluate the dimension of the mesh according to the accuracy needed and the dimensions of the room. The next Section deals with the experimental verification of these results.

**Table 2.** Simulated positioning error results.

| Configuration | | A Large | B Medium | C Tiny |
|---|---|---|---|---|
| **Full Room** | **Mean Error** | 0.71 m | 0.99 m | 3.11 m |
| | **Coverage \*** | 78.26% | 66.52% | 26.30% |
| **Within the Mesh Area** | **Mean Error** | 0.73 m | 0.51 m | 0.13 m |
| | **Coverage \*** | 73.21% | 93.81% | 100% |

**\*** Percentage of the full room area with sub-metrical localization error (≤1 m).

### 4.3. Experimental Results

The constellation of the 4 anchors network was implemented in the real case scenario depicted in Figure 9c. The three mesh configurations represented in Figure 9b define three different case studies. In the experiments described herein after, the mobile tag was installed on a tripod at a fixed height of 1.1 m and moved across the reference points indicated on the map of Figure 9a, while a convenient number of 5 RSSI traces were acquired and averaged for each point. These traces were exploited to estimate the mobile tag position by the estimator of Equation (3). Finally, some mechanical positioning techniques (i.e., laser position measurements, and mobile tag antenna alignment) provide a better consistency of the measured data across the experiments. These experiments have a two-fold objective: the main consists in drawing conclusions about the trade-off between mesh size, the localization accuracy and room area; the second is the assessment about the reliability of the simulation presented in the previous section.

Figure 12 depicts the CDF of the position estimation by the measured RSSI data for the three cases of study, respectively. Additionally, in this case the results are differentiated into two sets, namely the position errors inside the meshes and those across the entire room area, with each case of study differently colored.

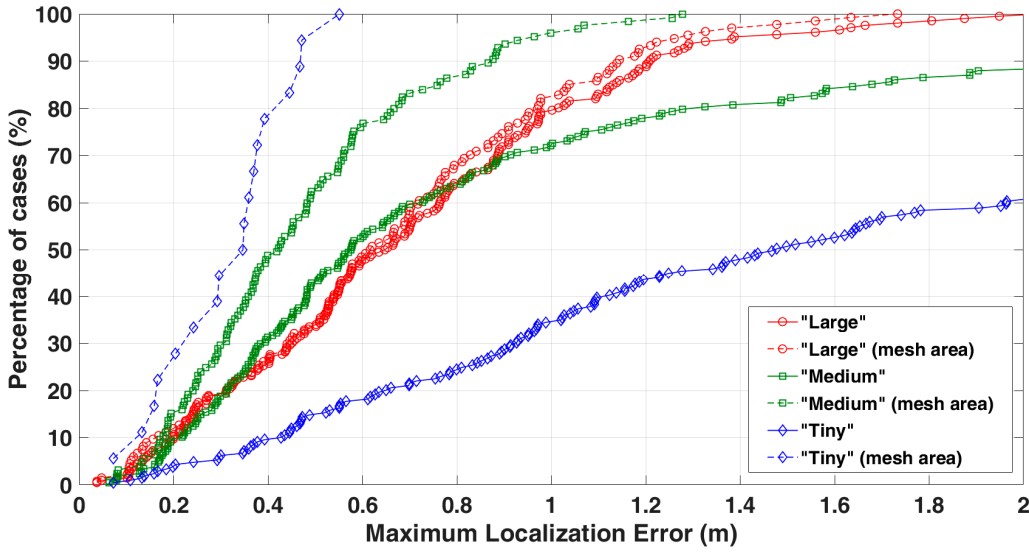

**Figure 12.** Measured localization error CDFs, for the three cases of study. Data acquired at the 299 points in the room of Figure 9a, assuming the mobile tag pointing upward, 5 averaged RSSI data for point and each antenna element, mobile tag height = 1.1 m.

This approach is motivated by the fact that for large areas a system architect could be interested in covering the domain by implementing a constellation of smaller meshes instead of adopting a larger one. This solution leads to better accuracy at the expense of increased system complexity. The localization results for the three experiments is summarized in Table 3.

**Table 3.** Measured localization error results.

| Configuration | | A<br>Large | B<br>Medium | C<br>Tiny |
|---|---|---|---|---|
| **Full Room** | **Mean Error** | 0.70 m | 0.85 m | 1.75 m |
| | **Coverage *** | 79.13% | 72.12% | 34.35% |
| **Within the Mesh Area** | **Mean Error** | 0.66 m | 0.47 m | 0.32 m |
| | **Coverage *** | 82.09% | 96% | 100% |

* Percentage of the full room area with sub-metrical localization error (≤1 m).

From the data we can observe what follows: Within the mesh boundary the localization accuracy increases when reducing the mesh area. Nevertheless, outside the same mesh the system is still able to provide effective localization with a meaningful accuracy. The wider mesh configuration, 'A', shows a good average localization error across the entire area, balancing its distribution all over the room. On the contrary, adopting the smallest mesh, i.e., 'C', the accuracy degrades significantly when evaluated in average across the entire room area. The 'B' mesh represents effectively a transition between the two cases, thus confirming the coherence between the experimental data.

It is noteworthy that CRB simulation results as shown in Figure 8 show a qualitative identical trend when compared with experiments. In comparison with the curves in Figure 8 the minimum achievable localization error within the mesh area shows a monotonous growth as the corresponding mesh size increases, exactly as depicted in Table 3. On the contrary, considering the entire room area, the average error decreases by increasing the mesh size until realizing an "optimum" mesh-size for which it is minimized.

The collected data leads to a straightforward conclusion: we can assess a tradeoff between network complexity and estimation accuracy; in fact, the smallest cell leads to the better local accuracy, while the larger leads to better mean error across the entire room. From this consideration, we foresee that considering smallest mesh sized repeated side-by-side lead to a better accuracy across larger areas, in place of one larger mesh. For example, following the data reported in Table 3 and considering the three mesh sizes, cf. Table 1, we see that the accuracy could be improved by a factor 2 (from 0.66 m to 0.32 m) moving from a single "large" mesh to a combination of about 20 "tiny" meshes, which would be necessary to cover the same area. Obviously, the significant increase of anchor nodes may cause a dramatic increase of costs, thus this trading-off drives the optimization of the positioning system between accuracy and complexity. Nevertheless, this assessment requires an experimental validation, which is at presently beyond the scope of this work.

Finally, from the comparison between the results provided in Tables 2 and 3, we can observe that for the cases of study 'A' and 'B' there are significant matches for both the 'full area' and 'within the mesh area'. The differences between the two accuracies are between 1 cm and 14 cm, and indicate that the measured results are consistent with the simulations of Section 4.2. Contrastingly, the comparison for the case of study 'C', i.e., the tiny mesh, reveals the worst agreement for both the data related to the 'entire room' and 'within the mesh area'. This is due to the effective difference between the mesh size and the room size, which amplifies the model inaccuracy given the fact that the small number of points within the mesh causes large relative fluctuation in the results.

*4.4. Discussion about the Simulated Positioning Results*

As emerges from the previous sections, the behavior of the localization accuracy allows to identify a trade-off for wider observation domains, where an overall acceptable accuracy is easily achievable without using a high number of anchors by exploiting larger meshes. By using a simple model like the one discussed in Section 4.2, the system architect can tune positioning accuracy by reducing their area and increasing the number of anchors, so that the localization system can be considered scalable for different application scenarios, [28]. This latter consideration can be sustained by comparing Figures 13 and 14, which show the maps of the localization error obtained with the three mesh dimensions;

the results come from simulations and measurements respectively, whose summaries are provided in Tables 2 and 3. The simulation model shows a significant level of agreement with the measured results, in particular for the large and medium meshes, for what concerns both the accuracy and the distribution of localization error. The accuracy of this model can be attributed to the adoption of circular polarized antenna that are inherently robust to multipath, [29,30]. This is due to the change of polarization rotation, i.e., left hand versus right hand, once that the electromagnetic wave bounces a conductive object, thus resulting in a cross-polarized wave at the receiver antenna that is therefore rejected. This occurrence justifies the adoption of a coarse channel model, like the one adopted in this model, although the adoption of more case of studies should be considered for a complete model validation.

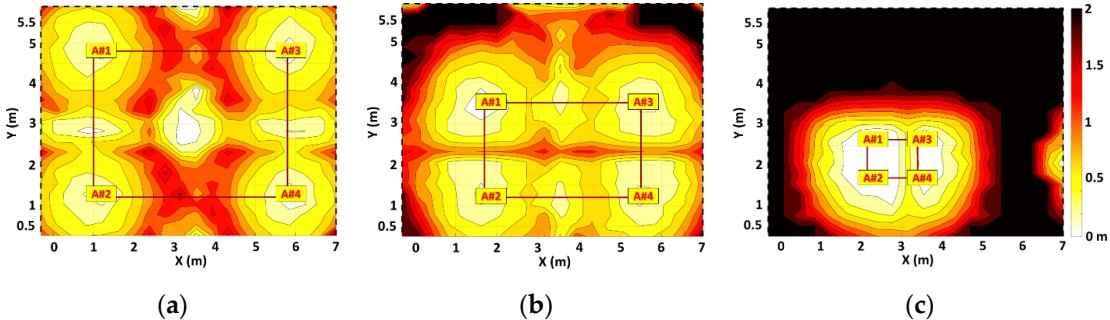

**Figure 13.** Simulated localization error distribution over room area for each mesh configuration: (**a**) large mesh; (**b**) medium mesh; (**c**) tiny mesh.

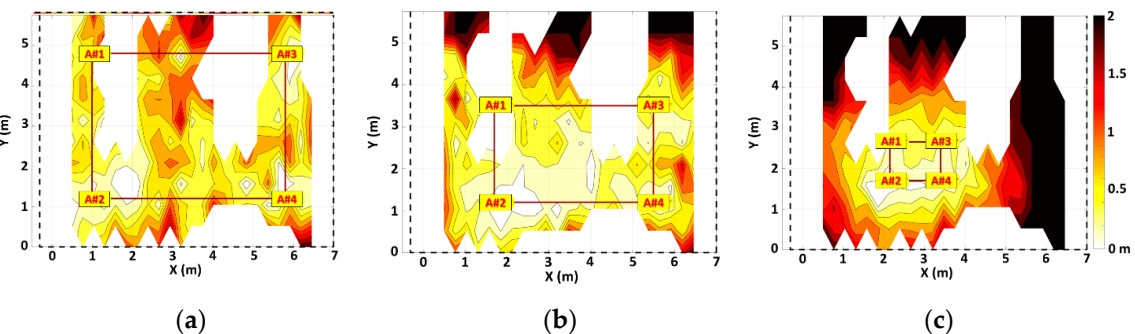

**Figure 14.** Measured localization error distribution over space for each mesh configuration (white area coincides with inaccessible room space, see Figure 9b), (**a**) large mesh; (**b**) medium mesh; (**c**) tiny mesh.

## 5. Conclusions

This paper has experimentally assessed the effects of the mesh area on localization accuracy for a distributed indoor localization system based on a constellation of four anchors composed of switched beam antennas and capable to acquire communication link RSSI, [15,16]. Due to its direct interoperability with an IEEE 802.11x/802.15.4 radio access technology, in prospective this system can be easily integrated in an existing wireless network, achieving the goal of providing a wireless data access system augmented with indoor localization, maintaining the same communication layers. This latter characteristic allows the WSN system architect designers to easily implement also smart routing network protocols, such as the one presented in [7].

The mesh area influences the localization accuracy significantly, demonstrating a clear trade-off between in-mesh and out-mesh results. The experimental data in the specific scenario presented in this work demonstrate that larger mesh provides a significant trade-off between accuracy and system complexity.

Provided that the aforementioned system is meant to allow an easy to integrate and non-invasive localization platform, the anchor mesh positions should be easily selected through an a priori process

that does not require complex steps of either calibration or testing. To help on this, the proposed simulation procedure has been validated by measurements results and thus can be used to forecast the optimal anchor network mesh area.

Table 4 shows a comparison between the presented system and actual state-of-art to the knowledge of the authors. Note that the system under consideration in this work is the only one that does not require any kind of training or supplemental data sources, and despite this its localization accuracy results are comparable with the one achieved by other state-of-art systems, while maintaining a similar density of anchors. This feature result is very important, and enabling for the optimization of future indoor localization system deployment.

**Table 4.** State of art comparison.

| Ref | N. of Anchor | Technology | Training | Room Size (m$^2$) | Anchor Density (1/m$^2$) | Mean Error (m) |
|---|---|---|---|---|---|---|
| [31] | 3 | CSI/RSSI | Yes | 40 | 0.075 | 0.60/0.90 |
| [32] | 4 | RSSI + IMU | Yes | 150 | 0.027 | 1.80 |
| [19] | 15 | RSSI + IMU | Yes | 1609 | 0.009 | 3.42 |
| [18] | 6 | RSSI + IMU | Yes | 157 | 0.038 | 1.00 |
| [33] | 6 | RSSI + IMU | Yes | 1250 | 0.005 | 1.00 |
| [34] | 4 | RSSI | Yes | 45 | 0.089 | 1.22 |
| [15] | 4 | SBA + RSSI | Yes | 34.5 | 0.120 | 1.08 |
| This Work | 4 | SBA + RSSI | No | 34.5 | 0.120 | 0.70 |

**Author Contributions:** Conceptualization, M.P. and A.C.; methodology, M.P. and S.M.; software, M.P.; validation, M.L.; formal analysis, M.P. and A.C.; investigation, M.P. and A.C.; resources, A.C. and G.C.; data analysis, M.L.; writing—original draft preparation, M.P.; writing—review and editing, M.L. and G.C.; writing—final version A.C.; visualization, M.L.; supervision, A.C.; project administration, G.P.; funding acquisition, G.C. All authors have read and agreed to the published version of the manuscript.

**Funding:** This research was funded by Regione Toscana POR CreO FESR 2014–2020—Action 1.1.5.a3—Bando FAR—FAS 2014, under the "SUMA—Struttura Urbana Multifunzionale Attiva" project.

**Conflicts of Interest:** The authors declare no conflict of interest. The funders had no role in the design of the study; in the collection, analyses, or interpretation of data; in the writing of the manuscript, or in the decision to publish the results.

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
