# Peer review of "Assessment of Anchors Constellation Features in RSSI-Based Indoor Positioning Systems for Smart Environments"

_electronics, doi:10.3390/electronics9061026_

Round 1
Reviewer 1 Report
This paper assesses the experimental results of the 4 anchor indoor positioning system. My main concern is as follows.
1, There are typos. To list a few:
smart environment paradigm, [1,2]. -> paradigm [1,2].
the internet itself, [3]. ->itself [3].
The ratio behind this approach -> reason
Received Signal Strength Indicator (RSSI)- received signal strength indicator
low-power -> low power
the a radio-frequency transceiver -> a radio-frequency transceiver
an height -> a height
and with a dihedral angle-> and a dihedral angle
Etc.
2, The paper lacks comparison with state-of-art technologies. Especially when the experimental results are not too impressive considering the actual very dense localization device deployment in a small room.
3, The experiment seems disconnected from the theory. The authors use the theory to demonstrate rectangular deployment should be better than the triangular one. Then the same theory can also be used to estimate the impact of changing mesh size, right? The authors should do a theoretical analysis of the impact of changing mesh size and then use experiments to validate the theory.
Author Response
Please find the reply to Review 1 in the attached file.

Reviewer 2 Report
Overall I am very pleased with your study. I believe it is very strong and the work is very sound.
My minor comments are:
- I suggest you clearly define in the beginning of your article what mesh area actually is, eg, area defined by the anchor placement (for those less familiar with 802.15.4 systems and jargon)
- I would clarify in the text why your CDF's do not reach 1
- Labels in Figure 7 are difficult to read and the caption falls on the top of the page 10
- On figure 8, I would add the X, Y values as a plot title or as another legend element
- The red text on figure C is hard to read
- On figure 11, I got a bit confused to see a Tiny line having such big localization error. While the text explains it sufficiently well, I would work out a bit the presentation on the figure, for example, grouping the mesh area lines together.
- On figure 11, I would prefer having shapes to help read the lines (I am color blind and it becomes hard to follow the middle lines)
- Your method scales quite well in existing networks where localization accuracy might need to be improved in certain facility areas.
- It would be interesting to see some figures on the power consumption of your tags and anchor devices
Author Response
please find the reply to Reviewer 2 into the attached file.
